# In Vitro and In Vivo Antiviral Activity of Gingerenone A on Influenza A Virus Is Mediated by Targeting Janus Kinase 2

**DOI:** 10.3390/v12101141

**Published:** 2020-10-08

**Authors:** Jiongjiong Wang, Richard A. Prinz, Xiufan Liu, Xiulong Xu

**Affiliations:** 1College of Veterinary Medicine, Yangzhou University, Yangzhou 225009, China; wjiongjiong@163.com; 2Department of Surgery, Northshore University HealthSystem, Evanston, IL 60201, USA; rprinz@northshore.org; 3Jiangsu Co-innovation Center for Prevention and Control of Important Animal Infectious Diseases and Zoonosis, Yangzhou University, Yangzhou 225009, China; xfliu@yzu.edu.cn; 4Animal Infectious Disease Laboratory, College of Veterinary Medicine, Yangzhou University, Yangzhou 225009, China; 5Institutes of Agricultural Science and Technology Development, Yangzhou University Joint International Research Laboratory of Agriculture and Agri-Product Safety, The Ministry of Education of China, Yangzhou University, Yangzhou 225009, China

**Keywords:** Gingerenone A, influenza A virus, JAK2, p70 S6 kinase, STAT3

## Abstract

Janus kinase (JAK) inhibitors have been developed as novel immunomodulatory drugs and primarily used for treating rheumatoid arthritis and other inflammatory diseases. Recent studies have suggested that this category of anti-inflammatory drugs could be potentially useful for the control of inflammation “storms” in respiratory virus infections. In addition to their role in regulating immune cell functions, JAK1 and JAK2 have been recently identified as crucial cellular factors involved in influenza A virus (IAV) replication and could be potentially targeted for antiviral therapy. Gingerenone A (Gin A) is a compound derived from ginger roots and a dual inhibitor of JAK2 and p70 S6 kinase (S6K1). Our present study aimed to determine the antiviral activity of Gin A on influenza A virus (IAV) and to understand its mechanisms of action. Here, we reported that Gin A suppressed the replication of three IAV subtypes (H1N1, H5N1, H9N2) in four cell lines. IAV replication was also inhibited by Ruxolitinib (Rux), a JAK inhibitor, but not by PF-4708671, an S6K1 inhibitor. JAK2 overexpression enhanced H5N1 virus replication and attenuated Gin A-mediated antiviral activity. In vivo experiments revealed that Gin A treatment suppressed IAV replication in the lungs of H5N1 virus-infected mice, alleviated their body weight loss, and prolonged their survival. Our study suggests that Gin A restricts IAV replication by inhibiting JAK2 activity; Gin A could be potentially useful for the control of influenza virus infections.

## 1. Introduction

Influenza is a highly contagious acute respiratory illness [1]. The disease is often presented as seasonal influenza epidemics and causes several hundred thousand deaths per year. In the United States alone, influenza accounts for an average of 36,000 deaths and >226,000 hospitalizations annually [1]. Four influenza pandemics in the past century, namely the H1N1 Spanish flu (1918), the H2N2 Asian flu (1957), the H3N2 Hong Kong flu (1968), and the H1N1 swine flu (2009), claimed millions of lives [2,3]. High morbidity and mortality, as well as the risk of an influenza pandemic, pose a major threat to public health [4]. Its causative agent, influenza virus, belongs to the *Orthomyxoviridae* family. Its genome contains eight negative-sense, single stranded RNA segments that encode 11 proteins. Influenza virus can be divided into the A, B, C, and D types that vary in host ranges and pathogenicity [5]. Influenza A virus (IAV) infects a wide range of avian and mammalian hosts, whereas influenza B virus infects humans and seals only [5]. Influenza C virus causes a mild respiratory infection and does not cause epidemics [5]. Influenza D virus primarily affects cattle and is not known to infect or cause illness in humans [5]. Based on a different combination of hemagglutinin (HA) and neuraminidase (NA), two viral surface glycoproteins, IAV can be further divided into many subtypes [1]. Several reassortant IAV genotypes such as H7N9, H5N6, H7N7, and H10N8 cause sporadic fatal infections in humans [6]. 

Vaccination and antiviral drugs such as M2 ion channel blockers and neuraminidase inhibitors are the mainstays of influenza prevention and treatment [4]. Favipiravir, also known as T-705, is an RNA polymerase inhibitor and has been approved for treating influenza virus infections in 2004 in Japan [7]. Baloxavir, a polymerase acidic (PA) protein inhibitor that binds to the PA endonuclease domain and blocks its cap-dependent endonuclease activity to cleave RNA, has been approved for treating influenza in several countries [8]. However, due to the lack of proofreading ability of the IAV RNA-dependent RNA polymerase, emerging IAV variants often become resistant to antiviral therapy, and vaccines lose their efficacy in protecting hosts from influenza virus infections [9]. There has been great interest in searching for the crucial cellular factors involved in virus replication and targeting them for antiviral therapy [4,10]. The NS1 protein of the H1N1 virus activates the PI-3 kinase pathway and inhibits the virus-induced apoptotic signaling responses to increase virus replication [11]. Targeting this pathway leads to the inhibition of IAV replication [12,13]. Recent studies using genome-wide screens to search for host factors as potential antiviral targets have led to the identification of a handful of molecules that play important roles in IAV replication [4]. Among them, Janus Kinase-1 (JAK1) and JAK2 are the leading drug target candidates whose deficiency profoundly dampens virus replication [14,15]. JAK inhibitors are capable of restricting IAV replication and have the potential to be developed as novel antiviral drugs. 

*Zingiber* species belong to the *Zingiberaceae* (ginger) family and have been widely used as spice additives and plant medicines [16]. Ginger possesses a variety of therapeutic effects, including anti-hyperglycemic, anti-thrombotic, anti-inflammatory, antioxidative, and radioprotective activities [17]. Ginger also exhibits various antimicrobial activities against viruses, bacteria, fungi, and nematodes [16]. Ginger extract restricts the replication of the herpes virus, rhinovirus, and respiratory syncytial virus [17,18]. The identity of compounds in ginger extract responsible for its antiviral activity remains unknown. Gingerenone A (Gin A), a compound extracted from ginger, is a dual inhibitor of S6K1 and JAK2 [19]. JAK1 and JAK2 have been recently identified as two crucial cellular factors implicated in IAV replication [14,15]. Our present study aims to determine the ability of Gin A to control virus replication and understand the mechanisms of action of Gin A on virus replication. Here, we report that Gin A was able to suppress the replication of three IAV subtypes in four different cell lines and in the lungs of IAV-infected mice; Gin A exerts its antiviral activity by inhibiting JAK2 activity. Our study suggests that Gin A could be potentially developed as a novel antiviral agent for the control of IAV infections.

## 2. Materials and Methods

### 2.1. Ethics Statement 

The experiments were approved by the Institutional Biosafety Committee of Yangzhou University. All experiments involving live virulent H5N1 viruses and animals were carried out in a P3-level biosafety lab certified by the Ministry of Agriculture, China. The protocols for the animal experiments were approved by the Jiangsu Administrative Committee for Laboratory Animals (approval number: SYXK-SU-2017-0007, 3 March, 2007), and complied with the guidelines of Jiangsu laboratory animal welfare and ethics of Jiangsu Administrative Committee of Laboratory Animals.

### 2.2. Reagents 

Ruxolitinib and PF-4708671 were purchased from Selleck Inc. (Shanghai, China). Gin A and antibodies against phospho-JAK2Y1007/Y1008 (sc-16566-R), β-actin (sc-47778), and GAPDH (sc-166574) were purchased from Santa Cruz Biotechnology, Inc. (Santa Cruz, CA, USA). Antibodies against phospho-tyrosine (P-Y-1000, #8954), STAT3 (#9139), phospho-STAT3Y705 (#9145), JAK2 (#3230), AKT (#4691), phospho-AKTS473 (#4060), S6K1 (#2708), phospho-S6K1T389 (#9234), S6 (#2217), and phospho-S6S235/236 (#4858) were purchased from Cell Signaling Technology (Danvers, MA, USA). An anti-nucleoprotein (NP) polyclonal antibody was prepared by immunizing mice with purified recombinant NP protein. An anti-M1 monoclonal antibody was prepared in our lab. An anti-PB2 antibody (#GTX125926) was purchased from GeneTex Inc. (Irvine, CA, USA).

### 2.3. Cell Culture and Viruses

Madin–Darby canine kidney (MDCK), A549 (a human lung cancer cell line of alveolar epithelial cell origin), 293T (a human embryonic kidney cell line), and DF1 (a chicken fibroblast cell line) cells were purchased from the American Tissue Culture Collection (Manassas, VA, USA). Cells were grown in DMEM containing 10% fetal bovine serum (FBS). The A/mallard/Huadong/S/2005 H5N1 virus (SY strain), a moderately pathogenic avian influenza strain, was isolated from poultry [20]. The virus was plaque purified three times in MDCK cells. A/PR8/34 (PR8) virus, a murine-adapted H1N1 subtype, was kindly provided by Dr. Liqian Zhu, Yangzhou University. A/California/04/09 (CA09), a pandemic H1N1 subtype of swine-origin, was kindly provided by Dr. Jinhua Liu, China Agricultural University. Avian Ck/SH/F/98 (F98) virus, an H9N2 virus subtype has been reported previously [21]. All IAV strains were propagated in 10 day-old specific-pathogen-free embryonic chicken eggs. Virus titers were measured by a 10-fold serial dilution (10^1^ to 10^9^), and each dilution (10^5^–10^9^) was used to infect chicken embryonic fibroblast (CEF) or MDCK cells. The 50% tissue culture infection dose (TCID_50_/mL) was calculated according to the Reed and Muench method. The IC_50_ values of Gin A required to inhibit IAV replication were calculated by using GraphPad Prism software (Version 8.0.0 for Windows, GraphPad Software, San Diego, CA, USA).

### 2.4. Virus Growth

To determine the effect of Gin A on virus growth, four cell lines (293T, A549, MDCK, and DF1) were infected with 0.01 multiplicity of infection (MOI) of H1N1, H5N1, or H9N2 virus and then incubated in the presence of various concentrations of Gin A (0, 10, 25, 50 μM). For H1N1 and H9N2 virus infection, tosyl-phenylalanyl-chloromethyl-ketone-treated trypsin (Sigma, St. Louis, MO, USA) (2 μg/mL) was added into the media during and after virus infection. The conditioned media were collected at 18, 24, or 36 hours post-infection (hpi). The TCID_50_ values were determined on MDCK cells according to our previous publications [20]. Data represent the mean ± standard deviation (SD) of four independent experiments. 

### 2.5. RT-qPCR Analyses

The 293T cells seeded in 6-well plates were infected with 0.01 MOI H5N1 virus and then incubated in the absence or presence of Gin A (0, 10, 25, 50 μM) for 16 h. Total RNA was extracted from H5N1 virus-infected cells by using a Qiagen RNA extraction kit. Viral RNA levels were measured by using a one-step reverse-transcription FRET-PCR to amplify the M gene according to our recent publication [22]. 

### 2.6. Luciferase Mini-Gene Assay

The 293T cells were transfected with 100 ng of luciferase reporter plasmid (p-Luci, kindly provided by Dr. Jinhua Liu) and the mixture of pcDNA3.1(+) plasmid constructs expressing the polymerase subunits *PB2, PB1, PA*, and *NP* genes from the A/mallard/Huadong/S/2005 H5N1 virus (200 ng each), and 20 ng internal control Renilla plasmid using Polyfect Transfection Reagent (Qiagen, Germantown, MD, USA). After incubation for 24 h, the cells were treated with dimethyl sulphoxide (DMSO) (0.5%) or with the indicated concentrations of Gin A (0, 10, 25, 50 μM) for another 24 h. The cells were harvested and analyzed for luciferase activity by using the Dual-Luciferase Reporter Assay System (Promega, Madison, WI, USA).

### 2.7. Western Blotting

The 293T, DF1, MDCK, and A549 cells seeded in 6-well plates were infected with H5N1, H1N1, or H9N2 (0.1 MOI each), and then incubated in the absence or presence of the indicated concentrations of various inhibitors for 8 h in the media containing 1% FBS. The cells were harvested and lysed in NP-40 lysis buffer (50 mM Tris-HCl, pH 8.0, 150 mM NaCl, 1% NP-40, 5 mM EDTA, 5 mm EGTA, 1 mM NaF, and 2 mM sodium vanadate, the cocktail of protease inhibitors (1×) (Pierce Chemical Co., Rockford, IL, USA), 2 mM sodium pervanadate). Cell lysates were prepared and analyzed for the expression of viral proteins (PB2, NP, and M1) or cellular proteins with their specific antibodies, followed by horseradish peroxidase-conjugated goat anti-rabbit immunoglobulin G (IgG) and SuperSignal Western Pico-enhanced chemiluminescence substrate (Pierce Chemical Co., Rockford, IL, USA). The density of the bands was analyzed by using an NIH Image-J software and normalized by the arbitrary units of their corresponding total proteins or β-actin. Quantified results were presented as the mean ± standard deviation (SD) from three experiments in bar graphs. 

### 2.8. In Vivo Experiments

The use of animals was approved by the Institutional Animal Care and Use Committee of the College of Veterinary Medicine, Yangzhou University. Female C57BL/6 mice were purchased from the Center of Experimental Animals of the College of Veterinary Medicine, Yangzhou University. All mice were maintained on a 12 h light/dark cycle and housed in ventilated cages at an ambient temperature of 23 °C. Mice were fed ad libitum on a normal chow diet (NCD). Gin A was first dissolved in dimethyl sulfoxide (DMSO) and then diluted in polyethylene glycol (PEG) 200 (Tokyo Chemical Industry, Inc., Tokyo, Japan). Mice (8 weeks old) (5 mice per group) were treated with Gin A (20 mg/kg/day) by gavage. Twelve hours after the first treatment, mice under diethyl ether anesthesia were intranasally mock-infected with PBS or infected with the H5N1 virus (1 × 10^5^ pfu in 50 μL PBS) by instillation. The H5N1 virus is a moderately pathogenic strain in mice. The dose was determined based on a previous publication from our group [23]. Mice were then treated with the same dose daily for three or five days. Mice were sacrificed by CO_2_ inhalation. Lung tissues were homogenized in a radioimmunoprecipitation assay (RIPA) lysis buffer (Cell Signaling Technology, Danvers, MA, USA) for Western blot or in PBS for analyzing virus loads. To determine the therapeutic effect of Gin A, mice (10 mice per group) infected with H5N1 virus were treated daily with Gin A (5 mg/kg body weight in 20 μL PEG 200) by intranasal instillation for 7 days. To avoid physical irritation caused by gavage and poor drug absorbance in the gastrointestinal tract of the H5N1 virus-infected mice, Gin A was given intranasally at a low dose in this experiment, since the drug, administered this way, directly goes to the lung where the virus replicates. Mice in control groups were treated with PEG 200. Mice were monitored daily for body weights and survival for 2–3 weeks and were humanely sacrificed by CO2 inhalation when they became moribund or when the loss of body weight decreased by >25%. The body weights of a small fraction of mice dropped beyond 25% in the last day before they were sacrificed. The last-day data of these mice were not included. 

### 2.9. Janus Kinase 2 (JAK2) Transfection

The 293T cells were transiently transfected with the empty vector or pBABE-JAK2, a retroviral vector encoding JAK2 tagged with a fragment of yellow fluorescence protein (YFP) (kindly provided by Dr. Eric Chang, Baylor College of Medicine, Houston, TX, USA) [24]. After incubation for 36 h, the cells were left uninfected or infected with the indicated MOI of the H5N1 virus and then incubated for 12 h. Alternatively, the cells were infected with the H5N1 virus (0.1 MOI) and then incubated in the absence or presence of various concentrations of Gin A (0, 10, 25, 50 μM) for 12 h. Cell lysates were prepared and analyzed for JAK2 and STAT3 phosphorylation and the levels of viral NP and M1 proteins. The virus titers in the conditioned media were collected and analyzed for the TCID_50_ values. Relative virus titers were calculated as the percent of controls. The results represent the mean ± SD of three independent experiments. 

### 2.10. Cell Proliferation Assay and the Selective Index (S.I.) Calculation

To exclude the possibility that the inhibitory effect of Gin A on virus replication was due to inhibition of cell proliferation, 293T cells were incubated with various concentrations of Gin A for 12 h and then analyzed for cell proliferation by using a CellTiter-Glo kit (Promega, WI, USA). To determine how selectively Gin A inhibited virus replication versus its cytotoxicity in the cells infected with IAV, 293T, A549, MDCK, and DF-1 cells seeded in 96-well plates with an approximately 80% confluence were incubated in the presence of various concentrations (0, 12.5, 25, 50, 100, 200 μM) of Gin A for 24 h. Cell viability was measured by using a CellTiter-Glo kit as described above. The IC_50_ values for cytotoxicity (C.C.) were calculated. The S.I. values were calculated by dividing the C.C. values with the IC_50_ values of Gin A required to inhibit virus replication. 

### 2.11. Statistical Analysis

Differences in virus titers, viral mRNA levels, the arbitrary units of scanned Western blot band density, and cell proliferation index were statistically analyzed by using an unpaired Student *t* test. Differences in the body weights of untreated and Gin A-treated mice were analyzed using a repeated measures analysis of variance (ANOVA) test. Differences in the survival of untreated and Gin A-treated mice were statistically analyzed by using a Log-Rank test. A *p* value of < 0.05 was considered statistically significant. All statistics were analyzed with SigmaPlot 11 software (Systat Software, Inc., San Jose, CA, USA).

## 3. Results

### 3.1. Gin A Inhibits Influenza A Virus (IAV) Replication

We first assessed the ability of Gin A to inhibit IAV replication in 293T cells, a human embryonic kidney cell line. The antiviral activity of Gin A was largely studied on the avian A/mallard/Huadong/S/2005 (SY) strain, a zoonotic virus of the H5N1 subtype with modest pathogenicity. The 293T cells infected with H5N1 virus (0.01 MOI) were incubated in the absence or presence of the indicated concentrations of Gin A (10–50 μM) for 18, 24, or 36 h. As shown in Figure 1B, Gin A reduced the titers of the H5N1 virus in the conditioned media in a dose-dependent manner. The IC_50_ values of Gin A to inhibit H5N1 virus replication in 293 T cells are between 10.2 and 24.5 μM. Of note, the baseline titers of IAV at 4 hours post-inoculation (hpi) were below the limit of detection. 

The 293T cells are not a natural host cell line for IAV. IAV replicates in this cell line with intermediate efficiency. Here, we tested if Gin A suppressed virus replication in DF-1 cells, a chicken fibroblast cell line, and A549 and MDCK cells, two highly permissive mammalian cell lines. As shown in Figure 1C, Gin A decreased H5N1 virus titers in the conditioned media of virus-infected DF-1, A549, and MDCK cells after incubation for 16 h in a dose-dependent manner. The IC_50_ values of Gin A to inhibit H5N1 virus replication in DF-1, MDCK, and A549 cells are approximately 21.1, 12.7, and 20.7 μM, respectively. Gin A dose-dependently lowered the titers of H1N1 (Figure 1D) and H9N2 viruses (Figure 1E) in the conditioned media of virus-infected 293T cells after incubation for 12 h. The IC_50_ values of Gin A to inhibit H1N1 and H9N2 virus replication in 293T cells are approximately 10.2 and 12 μM, respectively. The Selective Index (S.I.) values of Gin A to inhibit H5N1, H1N1, and H9N2 virus replication in 293T cells are 15.6, 15.7, and 13.2, respectively. The S.I. values of Gin A to inhibit H5N1 virus replication in A549, MDCK, and DF1 cells are 5.5, 5.7, and 19.4, respectively. RT-PCR revealed that Gin A dose-dependently lowered the levels of the *M1* gene of the H5N1 virus (Figure 1F). Consistently, Gin A also significantly decreased viral RNA polymerase activity in 293T cells in a luciferase reporter assay (Figure 1G). We then determined if the antiviral activity of Gin A was due to its anti-proliferative or cytotoxic effects. The 293T, A549, MDCK, and DF1 cells incubated in the presence of the indicated concentrations of Gin A for 12 h did not display any significant cytotoxicity or retarded cell growth (Figure 1H).

### 3.2. Gin A Decreases Viral Protein Levels 

Then, we determined if Gin A also reduced the viral protein levels of three IAVs in four cell lines. As shown in Figure 2, Gin A dose-dependently reduced the levels of the NP and M1 proteins in the MDCK, 293T, A549, and DF-1 cells infected with three IAV subtypes. Notably, Gin A reduced the levels of the M1 protein more profoundly than that of NP protein. Gin A lowered the viral protein levels of three IAV subtypes almost equally in these four cell line lines. We also examined the antiviral activity of Gin A on the CA09 virus, a pandemic H1N1 strain of swine-origin. The virus titers of this strain in the conditioned media of the 293T cells were too low to be detected. Western blot analysis revealed that Gin A reduced the levels of the NP and M1 proteins of this strain as effectively as those of the H1N1 PR8 strain (Appendix A). Immunofluorescence staining revealed that the fluorescent signals of the viral HA, M1, and NP proteins in H5N1 virus-infected 293T cells in the presence of Gin A (25 μM) were significantly reduced (Appendix A). This suggests that the antiviral activity of Gin A is indeed due to the restriction of virus replication but not due to its cytotoxic effect on 293T cells. 

### 3.3. Gin A Inhibits JAK2 and p70 S6 Kinase (S6K1) Activity

Gin A is a dual inhibitor of JAK2 and S6K1 [19]. Here, we tested if Gin A also inhibited the activity of JAK2 and S6K1 in IAV-infected cells. We first examined the ability of Gin A to inhibit JAK2 and STAT3 tyrosine phosphorylation in H5N1 virus-infected 293 T cells. H5N1 virus infection did not significantly increase JAK2 tyrosine phosphorylation in 293T cells at 0.5 (Figure 3A) or 8 hpi (Figure 3B). Interestingly, STAT3 tyrosine phosphorylation was weakly increased at 0.5 hpi but slightly decreased at 8 hpi, a phenomenon consistent with the observation in a prior study [25]. Gin A dose-dependently inhibited JAK2 and STAT3 tyrosine phosphorylation in uninfected or H5N1 virus-infected 293T cells at 0.5 or 8 hpi (Figure 3A,B). Consistently, Gin A also inhibited JAK2 and STAT3 tyrosine phosphorylation in 293T cells infected with H1N1 (Figure 3C) or H9N2 (Figure 3D) virus at 8 hpi. Western blot analysis with an antibody against phospho-tyrosine revealed that Gin A inhibited the total protein tyrosine phosphorylation (Figure 3E). Consistent with two previous studies showing that S6K1 inhibitors induce the feedback activation of the PI-3 kinase pathway [26,27], Gin A decreased S6 phosphorylation but increased AKT phosphorylation in a dose-dependent manner (Figure 3F). Of note, Gin A should also increase S6K1 phosphorylation but instead inhibited S6K1 phosphorylation, an observation that was also made in our recent study, suggesting its ability to inhibit mTOR activity as well [27].

### 3.4. Inhibition of IAV Replication by a JAK Inhibitor but Not by an S6K1 Inhibitor

Recent studies have shown that JAK1 and JAK2 play important roles in IAV replication [14,28,29]. We hypothesized that Gin A may inhibit IAV replication by inhibiting JAK2 activity. We investigated if Ruxolitinib (Rux), a JAK inhibitor, also inhibited IAV replication. Ruxdose dependently reduced the levels of the M1 and NP proteins (Figure 4A) in the lysates of 293T cells infected with the H5N1 virus and decreased the virus titers in the conditioned media (Figure 4B). Rux dose-dependently inhibited JAK2 tyrosine phosphorylation and dramatically inhibited the phosphorylation of its substrate STAT3 (Figure 4A). Western blot analysis with an anti-pTyr (P-Y-1000) revealed that Rux inhibited the total protein tyrosine phosphorylation (Figure 4C). The 293T cells incubated in the presence of the indicated concentrations of Rux for 12 h did not exhibit any significant cytotoxicity or retarded cell growth (Figure 4D). 

If Gin A restricts IAV replication in part by inhibiting S6K1 activity, PF-4708671, an S6K1-specific inhibitor, should also inhibit IAV replication. As previously reported [26], PF-4708671 inhibited S6 phosphorylation but induced AKT and S6K1 phosphorylation (Figure 4E). PF-4708671 did not affect the levels of viral NP or M1 protein levels. This suggests that the inhibition of S6K1 activity does not contribute to the antiviral activity of Gin A.

### 3.5. JAK2 Overexpression Enhances IAV Replication and Attenuates Gin A-Mediated Antiviral Activity

Prior studies have shown that JAK knockdown downregulates IAV replication [14,15]. Gin A and Rux, two JAK inhibitors, were able to inhibit JAK activities [19,30]. We tested if JAK2 overexpression led to increased IAV replication. As shown in Figure 5A, H5N1 virus dose-dependently decreased STAT3 phosphorylation, an observation consistent with the finding in a previous study [25]. JAK2 overexpression in 293T cells dramatically increased JAK2 and STAT3 phosphorylation (Figure 5A). JAK2 and STAT3 phosphorylation was slightly decreased in IAV-infected cells in a dose-dependent manner (Figure 5A). JAK2 overexpression significantly increased the levels of viral NP and M1 proteins (Figure 5A). JAK2 overexpression increased the virus loads in the conditioned media of the JAK2-transfected cells infected with 0.01, 0.1, and 1 MOI by 209%, 114%, 95%, respectively (Figure 5B). Of note, the actual infectious titers used to normalize to 100% in the pcDNA3.1-transfected cells infected with 0.01, 0.1, and 1 MOI were 103.11, 105, 105.5, respectively (Figure 5B). We then determined if JAK2 overexpression compromised the ability of Gin A to inhibit virus replication. As shown in Figure 5C, Gin A dose-dependently inhibited STAT3 and JAK2 phosphorylation and lowered the levels of viral NP and M1 proteins in pcDNA3.1-transfected 293T cells. However, Gin A inhibited STAT3 and JAK2 phosphorylation and decreased the levels of viral NP and M1 proteins in JAK2-overexpressed 293T cells in a significantly lower magnitude than in pcDNA-transfected 293T cells (Figure 5C). The virus titers in untreated or Gin A-treated 293T cells overexpressing JAK2 were significantly higher than their corresponding counterparts in pcDNA3.1-transfected 293T cells (Figure 5D).

### 3.6. Gin A Inhibits IAV Replication In Vivo

Finally, we investigated the antiviral effect of Gin A in mice infected with the H5N1 virus. Gin A treatment significantly decreased the levels of PB2, NP, and M1 proteins, and lowered the virus titers in the lungs on day 3 (Figure 6A,C) and 5 (Figure 6B,D), compared to the untreated controls. As shown in Figure 6E, the body weights of the Gin A-treated mice were significantly heavier starting on day 3 than that of the untreated mice. Forty percent of H5N1 virus-infected C57BL/6 mice intranasally treated with Gin A survived (Figure 6F). In contrast, the untreated C57BL/6 mice all died within 12 days post-infection. The median survival time of Gin A-treated mice was significantly longer than that of the untreated mice (*p* < 0.01) (Figure 6F). We also examined the antiviral activity of Gin A on H1N1 (PR8) virus-infected mice. Gin A treatment significantly alleviated the loss of body weight, compared to the body weight in the control group (Appendix A). Twenty percent of the C57BL/6 mice treated with Gin A by gavage survived H1N1 infection (Appendix A). The median survival time of Gin A-treated mice was significantly longer than that of the untreated mice (*p* < 0.01). 

## 4. Discussion

Natural products, particularly those derived from edible vegetables, tend to be more tolerable to the human body and to have the low levels of side effects [31,32]. Our present study shows that Gin A, a compound enriched in ginger roots (which contains >447 μg Gin A per 100 g) [19], was able to inhibit the replication of three different IAV subtypes in four different cell lines. Mechanistic investigation revealed that Gin A exerted its antiviral activity by inhibiting JAK2 tyrosine kinase activity. In vivo experiments revealed that Gin A inhibited H5N1 virus replication in the lungs of H5N1 virus-infected mice, alleviated their weight loss, and prolonged their survival. Our study suggests that Gin A could be potentially developed as a novel antiviral agent for the treatment of IAV infections. 

Protein tyrosine kinases have been increasingly recognized as potential molecular targets for antiviral therapy. There has been great interest in repurposing clinically approved PTK inhibitors as potential antiviral drugs for treating lethal virus infection. For example, c-Abl1 tyrosine kinase is involved in the productive replication of the Ebola virus. The inhibitor of c-Abl1 such as nilotinib suppresses the replication of the Ebola virus [33]. JAK inhibitors have been sought as potential antiviral drugs for treating HIV infection in vitro and in animal models [34,35,36]. JAK1 and JAK2 have been recently identified as the leading cellular factors that could be potentially targeted for anti-IAV therapy. Using an RNA-based screening assay, Watanabe et al. [14] reported that JAK1 plays a crucial role in the IAV virion assembly and that JAK1 siRNA and Rux inhibit IAV growth in A549 cells [14]. Using a genome-wide clustered regularly interspaced short palindromic repeats (CRISPR)/Cas screening method, Han et al. [15] reported that JAK2 is involved in IAV replication. Whether JAK inhibitors control IAV infection in vivo remains unknown. Our study shows that Rux dose-dependently decreased viral M1 and NP protein levels (Figure 4A). The concentrations of Rux used in our study are comparable to that by Watanabe et al. [14] showing that the drug at the concentrations of 30–100 μM significantly inhibits IAV replication. Gin A is a dual inhibitor of JAK2 and S6K1 kinases [19,27,37]. This inhibits the activity of JAK2 with the IC_50_ value of approximately 10 μM [19]. Our present study confirmed the ability of Gin A to inhibit JAK2 autophosphorylation and the phosphorylation of its substrate STAT3. Gin A lowered the levels of viral NP and M1 proteins in the lysates of IAV-infected cells and the virus titers in the conditioned media of IAV-infected 293T cells. These observations collectively suggest that Gin A inhibits IAV replication by inhibiting JAK activity. 

How JAK1 and JAK2 promote IAV replication is incompletely understood. JAK1 binds multiple viral proteins and is required for vRNP incorporation into virions [14]. Wang et al. [29] reported that JAK2 phosphorylates M1 at Y132 by JAK2 and stimulates its binding with importin-α1 and promotes M1 nuclear translocation. AG490 and dasatinib, a specific inhibitor of JAK and Src kinase, respectively, inhibit M1 nuclear translocation and virus replication. Han et al. [15] reported that JAK2 silencing leads to the increased expression of antiviral genes such as IFN-β, MxA, IFIT1, Rig-1, and ISG15, suggesting that the inhibition of JAK2 may enhance the antiviral defense activity of host cells. Our present study showed that Gin A inhibited JAK2 activity in IAV-infected cells. We speculated that the antiviral activity of Gin A is likely mediated by interfering with virion assembly and by enhancing the antiviral immune response. It should be noted that, while the JAK-STAT pathway is implicated in playing a critical role in antiviral innate immunity, the inhibition of this pathway does not necessarily lead to increased virus replication. For example, Zhang et al. [38] recently reported that the inhibition of STAT1 phosphorylation by fludarabine does not enhance but rather suppresses the H5N1 virus replication. This decreases the production of proinflammatory cytokines such as IL-6, TNF-α, IP10, IFN-α, and IFN-β in A549 cells and in the lungs of H5N1 virus-infected mice. Consistently, Nicol et al. [39] reported that IFN-γ receptor deficiency leads to decreased IAV replication and to lower inflammatory cytokine production in *IFNGR*^-/-^ mice than in wild-type mice. 

STAT3 is a transcription factor that plays a central role in regulating inflammation [40]. STAT3 is phosphorylated by JAKs at tyrosine 705 by many cytokine and growth factor receptors such as IL-6, IL-10, and EGF receptors [40,41]. Recent studies have shown that STAT3 activation suppresses apoptosis by inducing Bcl-2 and Bcl-xL expression [42]. STAT3 promotes cancer stem cell self-renewal and metastasis and plays an important role in carcinogenesis [42]. Numerous studies suggest that STAT3 activation can promote virus replication [41]. For example, STAT3 activation by varicella-zoster (VZV) virus in human fibroblasts in vitro and in xenografts in vivo can enhance VZV replication and pathogenesis [43]. STAT3 is also activated by the hepatitis C virus (HCV) and promotes HCV replication [44]. IB-32, an inhibitor of STAT3, suppresses HCV replication in hepatocytes [45]. Whether STAT3 also regulates IAV replication remains controversial. Mahony et al. [46] reported that the suppression of STAT3 expression leads to increased IAV replication and the decreased expression of antiviral genes, including PKR, OAS2, MxB, and ISG15. This suggests that STAT3 activation can suppress IAV replication. In contrast, Liu et al. [47] reported that neither the overexpression of constitutively active STAT3 nor S31-201, a STAT3-specific inhibitor, significantly altered the H1N1 virus (A/WSN/33) replication in A549 cells. Moreover, Hui et al. [25] reported that the H1N1 virus (A/Hong Kong/54/98) more effectively inhibits STAT3 phosphorylation than the H5N1 virus (A/Vietnam/1203/04) in human alveolar epithelial cells. Consistent with this observation, we found that the H5N1 virus dose-dependently decreased STAT3 phosphorylation in 293T cells at 8 hpi (Figure 5A), albeit the virus infection transiently increased STAT3 phosphorylation immediately following virus inoculation, which was blocked by Gin A (Figure 3A). However, the overexpression of a constitutively active STAT3 did not affect H5N1 virus replication (Xu, X, unpublished observations). We speculate that Gin A may exert its antiviral activity independent of its inhibitory effect on STAT3 activation. 

Ginger is a medicinal plant used in ancient China and an ingredient in Ge-Gen-Tang, a traditional Chinese medicine prescribed for treating human respiratory tract infections [17]. The water extract of fresh ginger and Ge-Gen-Tang inhibit human respiratory syncytial virus replication in vitro [18,48,49]. Imanishi et al. [50] reported that ginger extract can activate macrophages to produce cytokines that repress IAV virus replication. Ginger essential oil blocks herpes simplex virus type 2 (HSV-2) replication in RC-37 cells [51,52] and has a viricidal effect on caprine alphaherpesvirus 1 [53]. Which chemical component in ginger extract accounts for its antiviral activities remains unknown. Ginger extract contains two main categories of bioactive ingredients, e.g., gingerols and shogaols. We and others recently reported that Gin A can sensitize insulin receptor signaling and improve glucose metabolism by activating AMPK and by suppressing S6K1 activity [27,37]. In addition to its anti-hyperglycemic effect, our present study showed that Gin A was able to inhibit IAV replication in vitro and in vivo. Although the PI-3 kinase pathway has been implicated in playing an important role in IAV replication [12,13], our present study using an S6K1-specific inhibitor shows that the inhibition of S6K1 activity did not lead to the suppression of IAV replication, suggesting that Gin A inhibits IAV replication independent of its inhibitory effect on S6K1 activity. 

In our study, we also determined the S.I. value to assess the specificity of Gin A on virus replication. We found that the S.I. values were modestly above 10 in 293T but below 10 in A549 and MDCK cells. We postulated that the S.I. values could be affected by multiple factors. For example, the dose of IAV used to infect cells will likely affect the IC_50_ value of a candidate drug. Many studies have used a low MOI of IAV (0.01 or 0.001), which will lead to a much higher S.I. value. In our study, we used a 10-fold higher MOI (0.1) of IAV to infect four cell lines. In addition, the length of time used to measure the CC_50_ values also affects the S.I. values. The CC_50_ values in many studies are calculated based on the cytotoxicity data in which cells are incubated with antiviral agents for 24 h. In addition, Byun et al. reported that Gin A selectively induced apoptosis in cancer cell lines but not in normal cells [19]. This suggests that Gin A, even with modest S.I. values, preferentially causes cytotoxicity in tumor cell lines such as A549 and MDCK cells but becomes less cytotoxic in non-cancerous cell lines such as 293T and DF-1 cells. Gin A may not damage normal respiratory epithelial cells in vivo. In support of this notion, the antiviral activity of Gin A was achieved in our mouse model without any noticeable side effects when the mice were intranasally treated with Gin A at the dose of 5 mg/day. Moreover, Gin A has also been administered to mice in an obese mouse model [37] and a tumor xenograft model [19] for long-term uses without causing significant side-effects. Finally, Gin A is a natural product extracted from ginger roots, which should give a relatively safe profile. Of note, whilst ginger roots have been used in ancient Chinese as herbal medicine for treating febrile diseases, the concentrations of Gin A in raw ginger roots remain unknown. It is also not clear if the antiviral effects of ginger are mediated by other constituents. 

In the past decade, JAK inhibitors have been developed as novel anti-inflammatory drugs for treating rheumatoid arthritis and other inflammatory diseases [54]. Several ongoing clinical trials are testing their anti-inflammatory effects in COVID-19 patients [55,56]. We recently reported that A77 1726, the active metabolite of the anti-rheumatoid arthritis drug leflunomide, is able to inhibit the replication of influenza A virus and porcine epidemic diarrhea virus by targeting JAK1 and JAK2 [57,58]. Interestingly, a recent small-scale clinical trial revealed that leflunomide is very effective in treating COVID-19 [59]. Our findings that Gin A functions as a JAK inhibitor to inhibit IAV replication suggest that Gin A may control influenza virus infections not only through its antiviral activity but also by damping inflammation. Although Gin A could be potentially developed as an adjunct antiviral agent for treating IAV infections, Gin A will likely not be as effective as conventional anti-IAV drugs, which target neuraminidase, M2, or the viral RNA polymerase. The second cautious note is that Gin A and other JAK inhibitors, which target protein tyrosine kinases of host cells, are likely to have more severe side-effects than conventional antiviral drugs. Moreover, other clinically approved JAK-specific inhibitors may substitute Gin A once their antiviral effects are confirmed in animal models. 

In summary, our present study provides evidence that Gin A inhibits the replication of three IAV subtypes in vitro and controls IAV virus infection in vivo. Our mechanistic study suggests that Gin A suppresses IAV replication by inhibiting JAK2 activity. Gin A and other JAK inhibitors could be potentially developed as novel antiviral agents for the control of IAV infection in humans. Our study has validated the antiviral activity of a specific compound from ginger, a medicinal plant with antimicrobial activity.

## Figures and Tables

**Figure 1 viruses-12-01141-f001:**
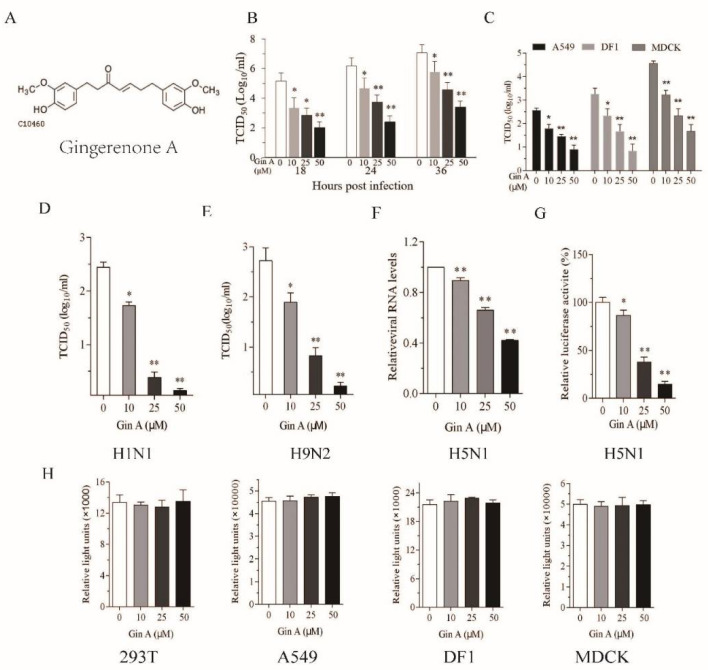
Gingerenone A (Gin A) inhibits influenza A virus (IAV) replication. (**A**) The chemical structure of Gin A; (**B**,**C**) The 293T (**B**), DF1, Madin–Darby canine kidney (MDCK), and A549 (**C**) cells were infected with H5N1 (0.01 multiplicity of infection, MOI) and then incubated in the absence or presence of the indicated concentrations of Gin A for the indicated lengths of time (**B**) or 16 h (**C**); The 293T cells were infected with the H1N1 (**D**) or H9N2 (**E**) virus (0.01 MOI each) and then incubated in the absence or presence of indicated concentrations of Gin A for 16 h. The virus titers in the conditioned media were analyzed by measuring the 50% tissue culture infection dose (TCID_50_) values; (**F**) Gin A reduces vRNA levels. The 293T cells infected with H5N1 virus were incubated in the absence or presence of the indicated concentrations of Gin A for 16 h. The total RNA was extracted and analyzed for the levels of the *M1* gene by real-time RT-PCR; (**G**) Gin A suppresses virus gene replication. The 293T cells were first transfected with the luciferase reporter genes plus the expression vector encoding *polymerase basic 1 (PB1), PB2, polymerase acidic (PA), and nucleoprotein (NP ) genes*. After incubation for 24 h, the cells were incubated in the absence or presence of the indicated concentrations of Gin A for another 24 h. Cells were harvested and analyzed for luciferase activity in a plate reader. The results represent the mean ± SD from one of three experiments with similar results, each was done in triplicate; (**H**) Gin A has little effect on 293T cell proliferation. The 293T cells seeded in 96-well plates were incubated in the absence or presence of the indicated concentrations of Gin A for 12 h. Cell proliferation was analyzed as described in the Materials and Methods. Data are the mean ± SD of three experiments. * *p* < 0.05; ** *p* < 0.01, compared to the untreated controls. Then, we determined if Gin A also reduced the viral protein levels of three IAVs in four cell lines.

**Figure 2 viruses-12-01141-f002:**
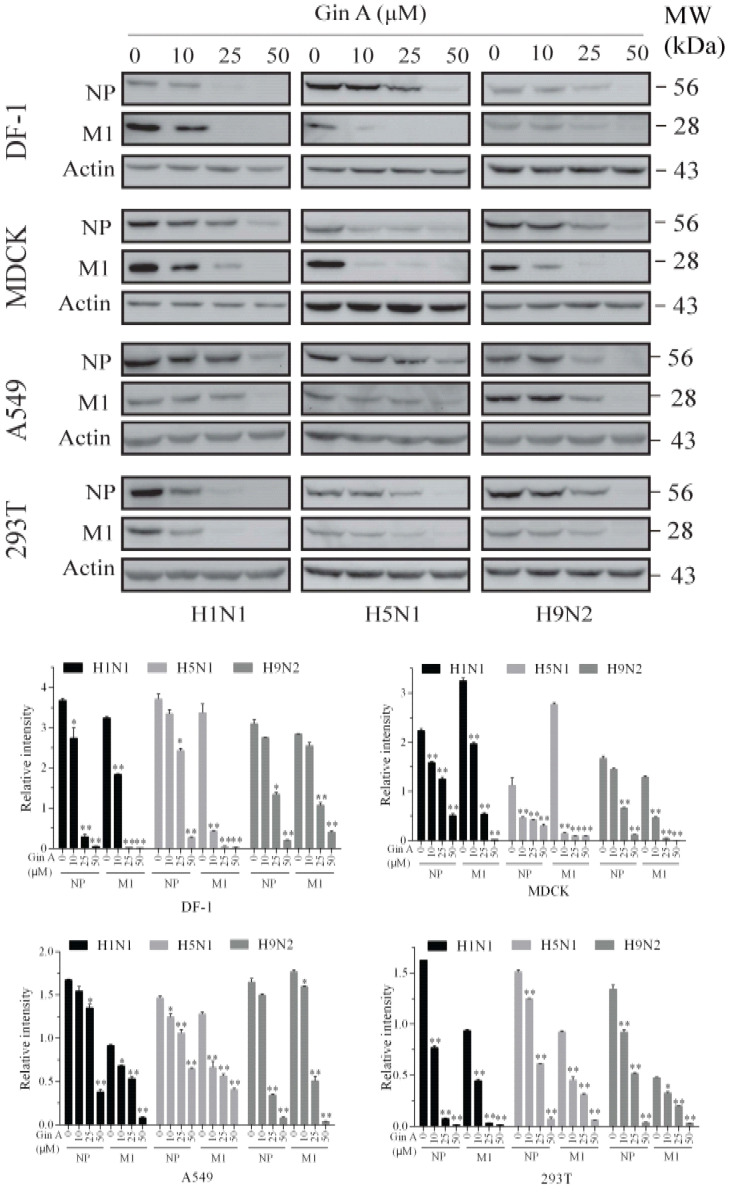
Gin A decreases viral protein levels. The DF-1, MDCK, A549, and 293T cells infected with the H1N1, H5N1, or H9N2 virus (0.1 MOI each) were incubated in the absence or presence of the indicated concentrations of Gin A for 8 h. Cell lysates were prepared and analyzed for NP and M1 expression by Western blot with their specific antibodies. Actin was detected as the loading control. The density of the viral protein bands was analyzed by using NIH Image-J software and normalized by the arbitrary units of β-actin. MW, molecular weight. * *p* < 0.05, ** *p* < 0.01, compared to the untreated control.

**Figure 3 viruses-12-01141-f003:**
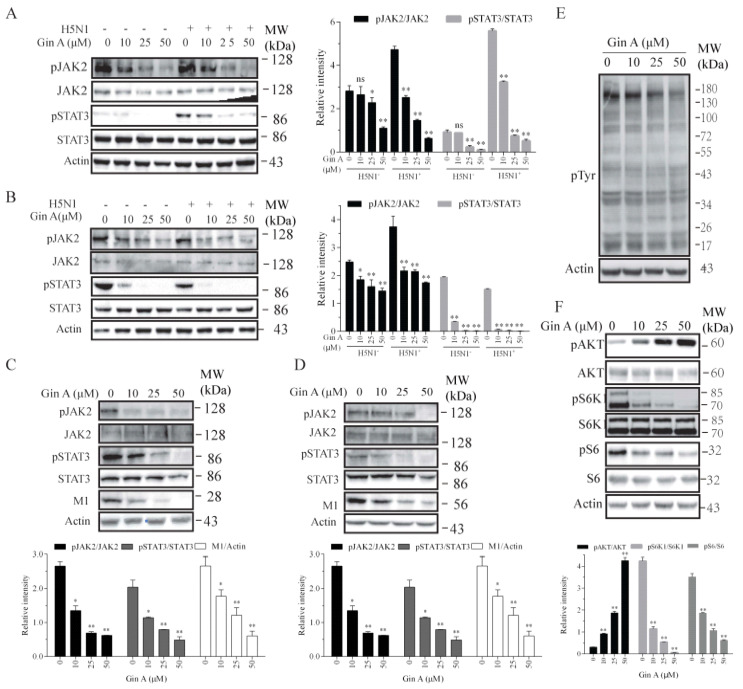
Gin A inhibits JAK2 and STAT3 tyrosine phosphorylation and S6K1 activity. The 293T cells were pretreated with the indicated concentrations of Gin A for 2 h. The cells were then infected with the H5N1 virus (1 MOI) at 4 °C for 2 h in the absence or presence of the same concentrations of Gin A. After removing the cells with warm PBS, the cells were incubated at 37 °C for 30 min in the absence or presence of the same concentrations of Gin A. The cell lysates were prepared and analyzed for JAK2 and STAT3 phosphorylation (**A**). (**B**–**F**) The 293T cells infected with H5N1 (**B**,**E**,**F**), H1N1 (**C**), or H9N2 (**D**) virus were incubated in the absence or presence of the indicated concentrations of Gin A for 8 h. Cell lysates were prepared and analyzed for JAK2 and STAT3 tyrosine phosphorylation and the levels of M1 expression by Western blot (**B**–**D**), for total protein tyrosine phosphorylation (**E**), and for the levels of protein phosphorylation in the PI-3 kinase pathway (**F**). Actin was detected as loading controls. The density of the phosphorylated JAK2 and STAT3 bands was analyzed by using NIH Image-J software and normalized by the arbitrary units of their corresponding total proteins. * *p* < 0.05, ** *p* < 0.01, compared to the untreated control.

**Figure 4 viruses-12-01141-f004:**
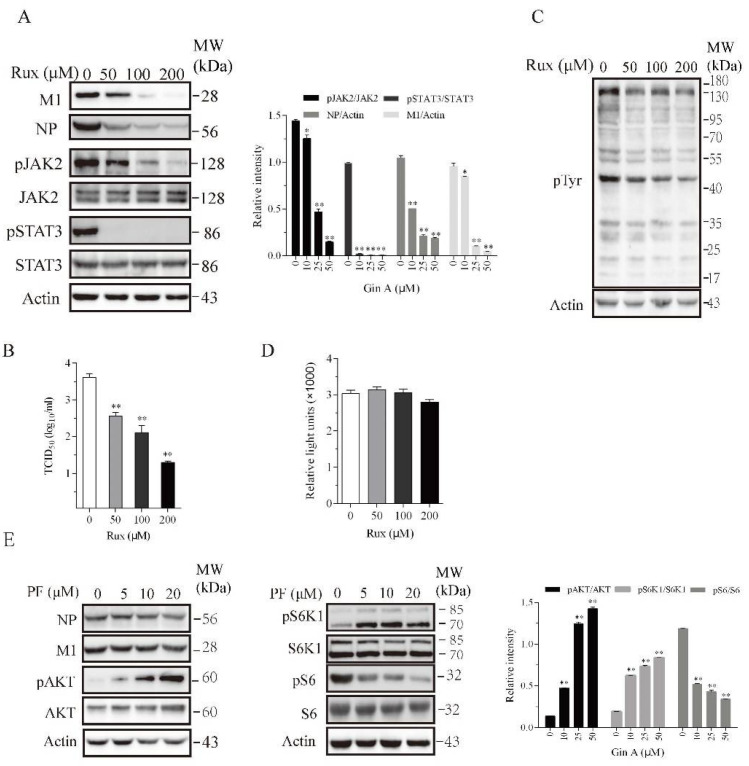
Decrease in H5N1 viral protein levels by Ruxolitinib but not by PF-4708671. The 293T cells were infected with the H5N1 virus (0.1 MOI) and then incubated in the absence or presence of the indicated concentrations of Ruxolitinib (Rux) (**A**,**C**) or PF-4708671 (**E**) for 8 h. Cell lysates were analyzed for the indicated proteins with their specific antibodies (**A**,**E**) or for the total protein tyrosine phosphorylation with an anti-pTyr mAb (P-Y-1000) (**C**). The density of the phosphorylated JAK2 and STAT3 or viral protein bands was analyzed by using NIH Image-J software and normalized by the arbitrary units of their corresponding total proteins or β-actin. * *p* < 0.05, ** *p* < 0.01, compared to the untreated control. (**B**) The virus titers in the conditioned media of 293T cells infected with H5N1 virus (0.1 MOI) at 8 hpi were measured by determining the TCID_50_ values. (**D**) Rux does not affect 293T cell proliferation. The 293T cells seeded in 96-well plates were incubated in the absence or presence of the indicated concentrations of Rux for 12 h. Cell proliferation was analyzed as described in the Materials and Methods. Data are the mean ± SD of three experiments. * *p* < 0.05; ** *p* < 0.01, compared to the untreated controls.

**Figure 5 viruses-12-01141-f005:**
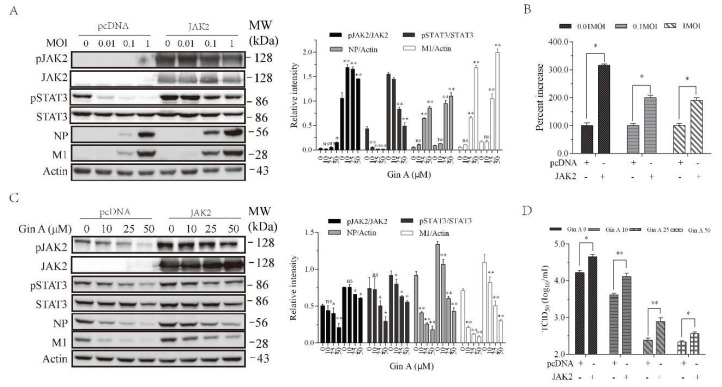
JAK2 overexpression attenuates the antiviral activity of Gin A. (**A**,**B**) JAK2 overexpression enhances IAV replication. The 293T cells were transiently transfected with the empty pcDNA3.1 vector or the vector encoding JAK2. After incubation for 36 h, the cells were left uninfected or infected with the indicated MOI of the H5N1 virus and then incubated for 12 h. Cell lysates were prepared and analyzed for JAK2 and STAT3 phosphorylation and the levels of viral NP and M1 proteins (**A**). The density of the phosphorylated JAK2 and STAT3 or viral protein bands was analyzed by using NIH Image-J software and normalized by the arbitrary units of their corresponding total proteins or β-actin. * *p* < 0.05, ** *p* < 0.01, compared to the untreated control. The virus titers in the conditioned media were analyzed for the TCID_50_ values (**B**). Relative virus titers were calculated as the percent of the controls; (**C**,**D**) The 293T cells were transiently transfected with the empty vector or the vector encoding JAK2. After incubation for 36 h, the cells were infected with H5N1 virus (0.1 MOI) and then incubated for 12 h in the absence or presence of the indicated concentrations of Gin A. Cell lysates were prepared and analyzed for JAK2 and STAT3 phosphorylation and for the levels of viral NP and M1 proteins (**C**). The virus titers in the conditioned media were analyzed for the TCID_50_ values (**D**). The results represent the mean ± SD of three independent experiments. * *p* < 0.05; ** *p* < 0.01. (**A**,**C**) The density of the phosphorylated JAK2 and STAT3 or viral protein bands was analyzed by using NIH Image-J software and normalized by the arbitrary units of their corresponding total proteins or β-actin. * *p* < 0.05, ** *p* < 0.01, compared to the untreated control.

**Figure 6 viruses-12-01141-f006:**
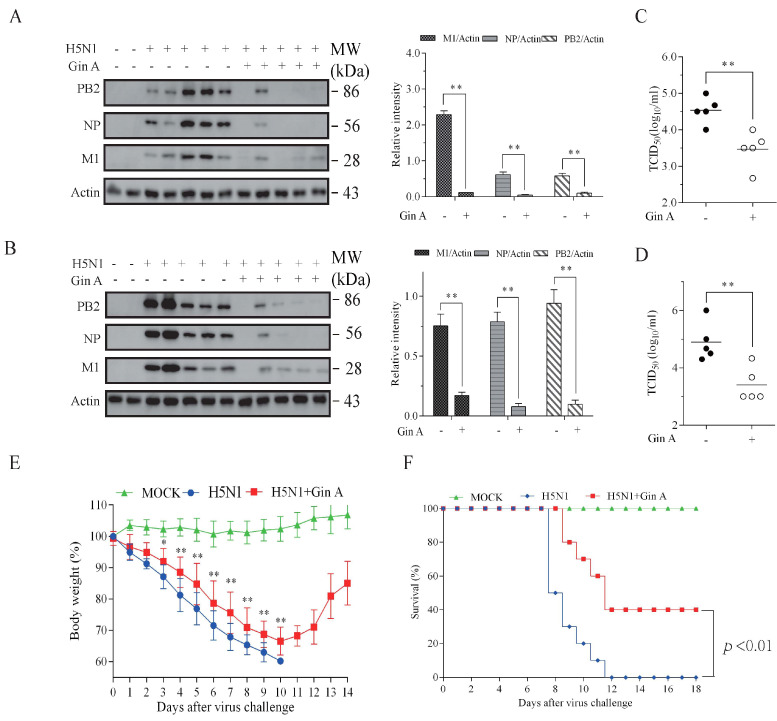
Gin A inhibits H5N1 virus replication in vivo. Female C57BL/6 (6–8 w old) mice were pretreated with Gin A 12 h before infection. Mice were intranasally infected with H5N1 virus (1 × 10^5^ pfu) and treated with the vehicle (polyethylene glycol (PEG) 200) or with Gin A (20 mg/kg body weight) by gavage daily for 3 (**A**) or 5 (**B**) days. Lung tissues were collected and analyzed for the levels of viral proteins (**A**,**B**) by Western blot and virus titers (**C**,**D**) by measuring the TCID_50_ values. PB2, NP, and M1 protein levels were determined by analyzing the band density of these proteins and then normalized by β-actin protein band density with NIH Image-J software. Each lane represents a lung tissue sample from an individual mouse. Data represent the mean ± SD of the lung tissues from five animals. (**E**,**F**) Mice (10 per group) were first treated with Gin A (5 mg/kg body weight in 20 μL polyethylene glycol (PEG) 200) 12 h before virus infection by intranasal instillation. After H5N1 virus infection (1 × 10^5^ pfu), the mice were treated daily with the same dose for 7 days. Mice were weighed and monitored for survival for 2 weeks. Body weights (**E**) and percent survival (**F**) were plotted. * *p* < 0.05; ** *p* < 0.01, compared to the untreated controls.

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
