# Peer review of "In Vitro and In Vivo Antiviral Activity of Gingerenone A on Influenza A Virus Is Mediated by Targeting Janus Kinase 2"

_viruses, 2020, doi:10.3390/v12101141_

Round 1

Reviewer 1 Report

Overall the article is well written and organized, except a few minor concerns:

  1. Line 69: should be drug target candidate;
  2. Line 117: (101 to 109) 1 and 9, (105–109) 5 and 9 should be superscript;
  3. Line 118: TCID50, 50 should be subscript;
  4. Line 119: IC50, 50 should be subscript;
  5. Line 132: or 16 hr should be for 16 hr;
  6. Line 167: 1X105, 5 should be superscript;
  7. Line 170 and line 179: CO2, 2 should be subscript;
  8. Line 201 and Line 202: IC50, 50 should be subscript;
  9. Figure 1B, right panel, move "Gin A (uM)" as X-axis label;
  10. Line 336: first sentence suppose to be "PF4708671 did not affect viral NP or M1 protein levels"? 
  11. Figure 6 legend, in vivo should be italic;
  12. line 392 and line 394: delete hpi;
  13. Line 424, should be 30-300 uM;
  14. RT-qPCR experiment didn't specify total RNA was extracted from infected cells or the supernatant? If it is from supernatant, what is the internal standard? 

Author Response

Comments and Suggestions for Authors:

Overall the article is well written and organized, except a few minor concerns:

Response:  many thanks to the reviewer for carefully reading our manuscript and spotting those typo errors. We have corrected all of them.

  1. Line 69: should be drug target candidate;

Response:  We replaced “drug candidates” with “drug target candidates” (drug).

  1. Line 117: (101 to 109) 1 and 9, (105–109) 5 and 9 should be superscript;
  2. Line 118: TCID50, 50 should be subscript;
  3. Line 119: IC50, 50 should be subscript;
  4. Line 132: or 16 hr should be for 16 hr;
  5. Line 167: 1X105, 5 should be superscript;
  6. Line 170 and line 179: CO2, 2 should be subscript;
  7. Line 201 and Line 202: IC50, 50 should be subscript;
  8. Figure 1B, right panel, move "Gin A (uM)" as X-axis label;

Responseï¼›We have replaced Fig. 1B

  1. Line 336: first sentence suppose to be "PF4708671did not affect viral NP or M1 protein levels"? 

Response:  Yes, we have corrected it.  Thanks!

  1. Figure 6 legend, in vivo should be italic;
  2. line 392 and line 394: delete hpi;
  3. Line 424, should be 30-300 uM;
  4. RT-qPCR experiment didn't specify total RNA was extracted from infected cells or the supernatant? If it is from supernatant, what is the internal standard? 

Response:  We have specified that we quantified viral RNA levels in infected cells.  

Reviewer 2 Report

The authors present the antiviral activity of Gingerenone A (Gin A) on influenza A virus in vitro and in vivo. The manuscript describes Gin A inhibited viral replication and decreased viral protein synthesis. Gin A showed antiviral effect on replication of three subtypes of IAV in four cell lines. Gin A also showed the inhibitory effect on JAK activity. The results are well presented and easy to follow.

  1. The manuscript described Gin A shows antiviral effect by inhibiting JAK2 activity. I feel that agents that target cell signalings have lower specificity of antiviral effects and are more likely to cause side effects than agents that directly target the viral proteins or viral components. Discussion about the difference of the target of antiviral agent would be helpful for reader.
  2. While the manuscript clearly showed Gin A suppress IAV replication via JAK2 inhibition, If inihibition of JAK is important for supression of IAV replication, JAK specific inhibitors should be used as antiviral agent. There is insufficient explanation about the benefits of using Gin A.
  3. While Gin A is a natural product from root of ginger, it is not described how high the concentration used in the manuscript compared to the concentration that exists in nature.

Author Response

Reviewer 2

Comments and Suggestions for Authors:

The authors present the antiviral activity of Gingerenone A (Gin A) on influenza A virus in vitro and in vivo. The manuscript describes Gin A inhibited viral replication and decreased viral protein synthesis. Gin A showed antiviral effect on replication of three subtypes of IAV in four cell lines. Gin A also showed the inhibitory effect on JAK activity. The results are well presented and easy to follow.

Response:  Many thanks to the reviewer for providing very thoughtful comments. 

  1. The manuscript described Gin A shows antiviral effect by inhibiting JAK2 activity. I feel that agents that target cell signaling have lower specificity of antiviral effects and are more likely to cause side effects than agents that directly target the viral proteins or viral components. Discussion about the difference of the target of antiviral agent would be helpful for reader.

Response:  We greatly appreciate that the reviewer brough up the issue of the specificity and potency of Gin A as an antiviral agent.  Gin A and other JAK inhibitors are not going to be as effective as conventional antiviral drugs and probably can only be used as an adjunct antiviral agent.  We added a paragraph reflecting the reviewer’s concerns (line 529-523)

  1. While the manuscript clearly showed Gin A suppresses IAV replication via JAK2 inhibition, If inhibition of JAK is important for suppression of IAV replication, JAK specific inhibitors should be used as antiviral agent. There is insufficient explanation about the benefits of using Gin A.

Response:  We share the reviewer’s concern that Gin A may not have an advantage over the existing JAK inhibitors.  It is not clear at this stage if Gin A is better than other JAK inhibitors.  Gin A is a natural product of ginger roots and a dual inhibitor of JAK2 and S6K1.  S6K1 is a serine/threonine kinase downstream of mTOR, a molecular target of immunosuppressive drug rapamycin.  Gin A may have better anti-inflammatory activity than the existing anti-inflammatory drugs.  We have added one sentence in the Discussion section (line 524-525)

  1. While Gin A is a natural product from root of ginger, it is not described how high the concentration used in the manuscript compared to the concentration that exists in nature.

Response:  We cited an article reporting that 447 mg of gingerenone A could be extracted from 100 g of ginger, the concentration in ginger roots is >125 mM (line 405).